# AIRSENSE-TO-ACT: A Concept Paper for COVID-19 Countermeasures Based on Artificial Intelligence Algorithms and Multi-Source Data Processing

Alessandro Sebastianelli [1] , Francesco Mauro [1], Gianluca Di Cosmo [1], Fabrizio Passarini [2] , Marco Carminati [3] and Silvia Liberata Ullo [1,*]

1   Engineering Department, University of Sannio, 82100 Benevento, Italy; sebastianelli@unisannio.it (A.S.); f.mauro@studenti.unisannio.it (F.M.); g.dicosmo@studenti.unisannio.it (G.D.C.)
2   Department of Industrial Chemistry Toso Montanari, University of Bologna, 40126 Bologna, Italy; fabrizio.passarini@unibo.it
3   Dipartimento di Elettronica, Informazione e Bioingegneria, Politecnico di Milano, 20133 Milan, Italy; marco1.carminati@polimi.it
*   Correspondence: ullo@unisannio.it

**Abstract:** The aim of this concept paper is the description of a new tool to support institutions in the implementation of targeted countermeasures, based on quantitative and multi-scale elements, for the fight and prevention of emergencies, such as the current COVID-19 pandemic. The tool is a cloud-based centralized system; a multi-user platform that relies on artificial intelligence (AI) algorithms for the processing of heterogeneous data, which can produce as an output the level of risk. The model includes a specific neural network which is first trained to learn the correlations between selected inputs, related to the case of interest: environmental variables (chemical–physical, such as meteorological), human activity (such as traffic and crowding), level of pollution (in particular the concentration of particulate matter) and epidemiological variables related to the evolution of the contagion. The tool realized in the first phase of the project will serve later both as a decision support system (DSS) with predictive capacity, when fed by the actual measured data, and as a simulation bench performing the tuning of certain input values, to identify which of them led to a decrease in the degree of risk. In this way, we aimed to design different scenarios to compare different restrictive strategies and the actual expected benefits, to adopt measures sized to the actual needs, adapted to the specific areas of analysis and useful for safeguarding human health; and we compared the economic and social impacts of the choices. Although ours is a concept paper, some preliminary analyses have been shown, and two different case studies are presented, whose results have highlighted a correlation between $NO_2$, mobility and COVID-19 data. However, given the complexity of the virus diffusion mechanism, linked to air pollutants but also to many other factors, these preliminary studies confirmed the need, on the one hand, to carry out more in-depth analyses, and on the other, to use AI algorithms to capture the hidden relationships among the huge amounts of data to process.

**Keywords:** COVID-19 counteractions; risk levels; artificial intelligence; long short term memory neural network; satellite remote sensing; sensor networks; pollutants; macroanalysis; microanalysis; air quality; environmental chemistry; anthropogenic activities

## 1. Introduction

COVID-19 pandemic started from China towards the end of 2019, and when the consequences of its spreading became clear, the Chinese Government immediately took actions to protect its citizenship by activating restrictive measures on mobility, and on industrial and commercial activities, before turning to a total lockdown for the area of Wuhan, which had been identified as the most affected area in the Hubei region. Many other

governments and local institutions (regions and municipalities) worldwide have applied severe lockdown measures, but have always made a posteriori decisions: increasing levels of lockdown have been activated, based on the number of infected, hospitalized and dead, all the up to a generalized lockdown, like the one imposed in Italy from the beginning of March until almost end of June, and in many other countries worldwide as well. These measures of generalized lockdown were adopted by imitating what was decided in the Wuhan area, to contain the spread of COVID-19, since they had proved to result in a positive impact in terms of the number of infections [1,2].

However, the lockdown measures have proven to result in different outcomes in different countries, since several factors impact the efficiency of the lockdown levels. It is important to evaluate these differences and understand why the dynamics are specific for each country. The habits and interaction modes of the populations, together with their abilities to respect the rules of social distancing, affect COVID-19's spread. With these characteristics, other factors must be taken into consideration to describe the responses to the lockdown measures and the reduction in the number of new infected, such as the population density. An interesting analysis has been presented in [3], where an interactive tool has been presented to monitor the trends of COVID-19, analyzed in terms of new infected for 187 countries. Many European countries have showed a rise–fall trajectory of two weeks, with Italy and France characterized by the fastest decrease over the last lockdown period at about the end of May and the first days of June. In [3] the mortality rate has also been plotted and discussed, defined as the number of deaths on the number of detected infected. Countries such as Germany and Austria have shown the smallest values with respect to other countries. Mortality rate appears to be very different among the countries, and this is mostly related to the different factors affecting this number: socio-economic conditions, genetic predisposition, sex, age [4], past health status [5], severity of the virus [6], efficiency of hospital services and availability of intensive care places [7]. Clearly, the mortality rate is a number affected by the values of deaths and infected, as reported in the available public databases. Therefore, some aspects affect the values, such as people dead because of COVID-19 but not reported so, and infected people not detected because of being asymptomatic or because of not being swabbed. One thing has appeared clear: the social distancing of the population was the main factor increasing or reducing the spread of the disease. Many studies have been carried out and models developed to analyze this phenomenon [8–12]. However, if the lockdown measures, as discussed, result on one hand in the reduction of infected due to the increasing of social distancing and isolation, on the other hand, these methods of intervention were not appropriate due to their negative implications for economic [13] and social aspects [14,15]. While the negative impact of the restrictions' measures on the economy appears obvious, a deeper analysis is necessary in the latter case. In particular, in [14] the affect of lockdowns on the increase of obesity in youths is shown, and in [15] the greater risk for damage to the people's health, physical and mental, is presented, all due to inactivity and the long stay at home.

Given all the above considerations, generalized lockdowns cannot be the solution, and future interventions must learn from the past to make a priori decisions, which have a minimal impact on commercial activities and social aspects, but are equally effective in limiting the spread of the virus.

Since huge amounts of data are involved in the process of COVID-19 spread, and their interactions are captured with difficulty through traditional methods, AI algorithms are suitable tools with which to capture the hidden interactions between data, and to provide the possibility for a micro and macroanalyses of the phenomenon, by allowing localized interventions.

In this article, we aim to present the concept of such a tool, based on AI algorithms, to help institutions and decision makers in adopting a priori targeted measures. Moreover, such a tool should be able to help also in monitoring the situation in real time, to avoid what happened unfortunately in many places in the world, and also in Italy, when the lockdowns

were done: after a few weeks, the situation started to worsen again, in terms of infected and hospitalized, and the previously adopted measures of lockdown were resumed.

Although ours is a concept paper, some preliminary analyses have been also shown and two different case studies presented, whose results have highlighted a correlation between $NO_2$, mobility and COVID-19 data, but given the complexity of the virus's diffusion mechanism, linked to air pollutants but also to many other factors, the need was confirmed on the one hand to carry out more in-depth analyses, and on the other to use AI algorithms to capture the hidden relationships between the huge amounts of related data. It is worth highlighting, as explained in detail ahead in this paper, that a preliminary analysis of some parameters has been carried out also to identify those indicators more efficient than others in training the neural network chosen for the proposed model.

To the best of our knowledge, this proposal represents significant progress compared to the state of the art. The only studies related to similar issues concern the creation of the ESA RACE dashboard [16], published on the 5 June 2020, which allows the use of satellite data to support the monitoring of commercial and productive activities, and on the other hand, the development of national and international projects, such as PULVIRUS [17], EpiCovAir and RESCOP [18], for the study of the correlation between suspended atmospheric particulate matter concentration and COVID-19. Our proposal integrates the above, and goes beyond their aims, by proposing a decision support system (DSS) capable of producing the level of risk, but also useful to be used for simulations, for tuning the input values to establish which inputs to vary and how to obtain a lower level of risk. Moreover, a particular note goes to the work presented in [19], where a theoretical framework for an early warning and adaptive response system (EWARS) was conceptualized. The proposed system was based on fuzzy logic for the processing of data from different sources and aimed to support decision makers in preventing and fighting epidemics. A new version of this model was presented in [20] ten years later by the same authors, where the employment of smartphone-based applications was discussed and a mosquito perception index (MPI) was introduced in order to manage the dengue infection, which affects many African and Asian countries. With respect to [19,20], the model described in our paper is different because it relies on a day-to-day use of freely available Sentinel-5P satellite data, besides other data, retrieved by public databases or collected by ground-based sensor networks. Moreover, in our model we include a long short term memory (LSTM) neural network, instead of fuzzy logic-based algorithms, for data processing.

In short, the main contributions of our work are as follows:

- The description of a new tool based on AI algorithms to support institutions in the implementation of targeted countermeasures against emergencies, such as the current COVID-19 pandemic.
- Some preliminary analyses showing the correlations amoung $NO_2$, mobility and COVID-19 data.
- The confirmation about the need to use AI-based tools to capture hidden relationships among data not detectable through traditional methods.

The manuscript is organized as follows: In Section 2, the DSS, able to suggest a priori interventions, is summarily introduced, and the main factors involved in the DSS design are described; Section 3 presents the proposed architecture in more detail; a preliminary analysis of the data used to feed the model has been introduced in Section 4; Section 5 presents the chosen case studies where the correlation between $NO_2$ and new cases of COVID-19 is analyzed, also using mobility data; Section 6 is dedicated to a discussion of the above findings, by trying to highlight the main contributions. Conclusions are highlighted in Section 6, along with future work.

## 2. A Decision Support System to Face COVID-19

In this concept paper, we aim at focusing on the challenges posed by the COVID-19 pandemic and present a multidisciplinary and quantitative approach to respond to the sanitary and economic crisis. The proposed model is named AIRSENSE-TO-ACT, and

is the description of a project submitted to the 2020 FISR (Fund Integrated Special for Research) Call of MIUR (Italian Ministry of University and Research), issued to collect solutions related to the diffusion of the COVID-19 pandemic, able to contain its effects and offering a novel way for the management of the reorganization of activities and processes.

The model is based on the fusion of heterogeneous data coming from different sensors: on-board satellites and/or positioned on ground platforms, both mobile and fixed, in addition to other public data extracted from databases [16,21–23], all jointly processed through the application of Machine Learning (ML) algorithms [24–28], and the employment and comparison of macro and microsystems of analysis.

As regards the ML algorithms, previous works have highlighted their importance for applications related to COVID-19. In particular, [25,26] used a combination of ML-based algorithms for improving the COVID-19 diagnosis and helping the doctors to identify its presence in the available images, and [27] presented a forecasting model of COVID-19 spread based on LSTM networks. It uses LSTM networks for prediction, based on the past numbers of infectious diseases, and it compares the transmission rates between Canada and Italy, and USA countries. This research represents a demonstration of how well a LSTM network can work in learning from the past to produce later the desired output, and it supports the decision to choose a LSTM network for the tool proposed in our manuscript.

As regards the data, it is necessary to highlight that they can be indeed extracted from public databases (i.e., epidemiological information, number of infected, mobility information, satellite data, etc.), which have further expanded recently, as a consequence of the general awareness due to the COVID-19 emergency, but to improve the model and the analyses to be implemented, it is foreseen in our work the possibility to carry out data collection campaigns through ground-based networked sensors, for instance, to validate the developed model, in particular, in the Italian areas where the emergency has shown more critical issues, such as in Po Valley, and specifically in the Lombardy region. Local and punctual measures with better spatial and temporal resolution will increase the effectiveness of the fight against the spread of the contagion, allowing us to focus the analysis from macroareas covered by the satellites, to microareas monitored through local (mobile or fixed) sensor networks, in order to increase the "granularity" of the data and to reduce the reaction time of the decisions to be made.

It is necessary to underline that a crucial aspect will concern the creation of the datasets with which the artificial network underlying the DSS needs to be trained, since based on previous experience, this activity may take up to 3/4 months. This will be discussed in detail ahead in the manuscript, in Section 3.1.

The proposal falls mainly within the scope of risk prevention, developing solutions to counteract and contain the effects of COVID-19 and any future pandemics. However, thanks to the versatility of the proposed tool, it is also of immediate use for the response to other emergencies, to develop tailored solutions and for the management of the organization of activities and processes, relating to the phase of overcoming the phenomenon in safety conditions. Moreover, it is also worth pointing out that, although developed to combat the COVID-19 pandemic in Italy, this model can be applied to other emergency situations wherein air quality has health implications, above all in other countries with similar economic and infrastructural characteristics.

*2.1. Input Data and Analytical Tool*

Going into the detail, the proposed model, as already mentioned above, aims to combine satellite data, and data acquired through ground platforms, both mobile and fixed, related to several types of information, such as the concentrations of some pollutants like NOx, PM10 and PM2.5; meteorological data; air mass displacement; chemical–physical parameters, such as temperature and humidity; and other reference data, such as population mobility, epidemiological data (number of infected), number of places still available in intensive care (global values or per-hospitalization points, distributed throughout the regions, nationally, etc.), the concentration of residents per $km^2$, the degree of implemented

lockdown and so on. Several studies have already analyzed the impacts and correlations of the mentioned factors on COVID-19 [13,29–35]; moreover, the past experience on what happened has demonstrated that the listed factors each have a different influence on COVID-19, and their interactions can lead to even worse situations.

The DSS presented in this manuscript represents a tool able to produce as output the level of risk at micro and macrolevels, based on the correlation between the selected data. The choice to base the proposed model on AI algorithms stems, as highlighted before, from the enormous amount of data which must be taken into account, and above all from the need to capture the correlations, sometimes hidden, between this information. In fact, completely different causality structures emerge from the literature when the various factors are considered in combination. For example, the rise in temperature seems to lead to a reduction in the diffusion of COVID-19, but it depends also on the humidity values, since it has been found that high temperature values with high humidity values do not stop the spread of COVID, but on the contrary facilitate it. For instance, in [36] an index to evaluate the relative COVID-19 risk due to both weather and air pollution, the covid risk weather (CRW) parameter, has been introduced. The CRW can be used to compare the relative changes in reproductive number for the disease due to the weather factors (average and diurnal temperature, ultraviolet (UV) index, humidity, pressure, precipitation) and air pollutants ($SO_2$ and ozone). In [36] the authors highlighted that warmer temperature and moderate outdoor UV exposure may offer a modest reduction in reproductive numbers. However, UV exposure can not fully contain the transmission of COVID-19. If on one hand both high temperature and solar radiation are able to speed up the inactivation rate, on the other high relative humidity may promote the diffusion rate [36,37]. Still, in the literature on the diffusion of respiratory diseases, the interactions between multiple elements—pollution, high population density and overcrowding—have been analyzed, and recent studies have shown a correlation between high concentrations of fine particulate matter and COVID-19 diffusion [38,39]. However, the subject is still much debated, and further multidisciplinary investigations are in progress.

*2.2. Key Factors in the DSS Design*

Each country and its government has a department of civil protection managing the situations in case hazardous events happen. For instance, in Italy the Department of Civil Protection [40] activates specific interventions when several risks, such as seismic, volcanic, health, environmental, fire-related and others, occur. In the specific case of the COVID-19 pandemic, the problem and its solutions have involved three main acting factors:

- **Pollution and population**: pollution, overpopulation and anomalies in climate conditions can lead to the disease.
- **Spread of diseases**: the spread of diseases involves issues related to population and some geo-physical changes, but it can be mitigated by making correct decisions
- **Decisions made**: decisions are made by humans based on previous experience, but to make correct decisions the right information and robust DSSs are needed.

To move from an a posteriori model to an a priori model, the proposed tool has been designed to take into consideration multiple parameters from multiple sources, to produce as output the degree of risk. Its general scheme is represented in Figure 1, where human activity's contribution and economic activity is highlighted, but interactions with factors such as climate conditions, pollution density, mobility data, etc., result in the number of infected and other effects, whose mutual interactions can be captured through AI-based paradigms.

Clearly, such a system, to be effective, must rely on the availability of data almost in real time, or with a low revisit time (i.e., Sentinel-5P and others), in order to help in realizing a continuous monitoring system, able to produce the right alert to manage jointly multiple risks, such as in the case discussed, sanitary, environmental and economic risks.

Based on recent events and the studies of the researchers working on COVID-19 diffusion, it came out that there is a time lag between the contagion moment and the

manifestation of COVID-19 symptoms in the infected person (and its positive test results), a period evaluated in Italy as being between 14 and 21 days, but which can be very different in other countries, as reported in [3] and already discussed in the Introduction section. Similarly, the closure of road traffic and the COVID-19 lockdown restrictions may have an effect on the reduction of pollution, but this happens only after some days, and the elapsed time is different in different areas. The situation is even worse particularly in some parts of Italy, such as Po Valley, which suffers from weather and orographic conditions that make it very problematic to shuffling the air masses.

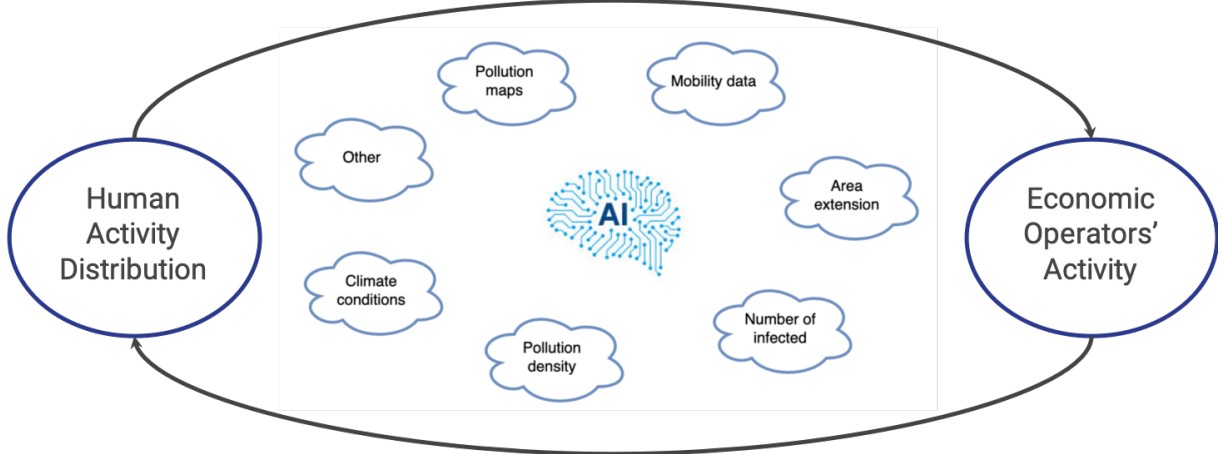

**Figure 1.** Data sources.

Given the above considerations, it appears obvious that the delayed interaction between causes and effects must be captured in the model.

Therefore, a particular network architecture called LSTM was chosen for the creation of the DSS, to process historical data series and predict not only the level of risk at the generic time T, but also in other subsequent instants, so that the network, after being trained, can be used as a transfer function to calculate the output (the expected risk level) by using the new input data. The proposed architecture and the LSTM network are shown in Figures 2 and 3, and described in the next section.

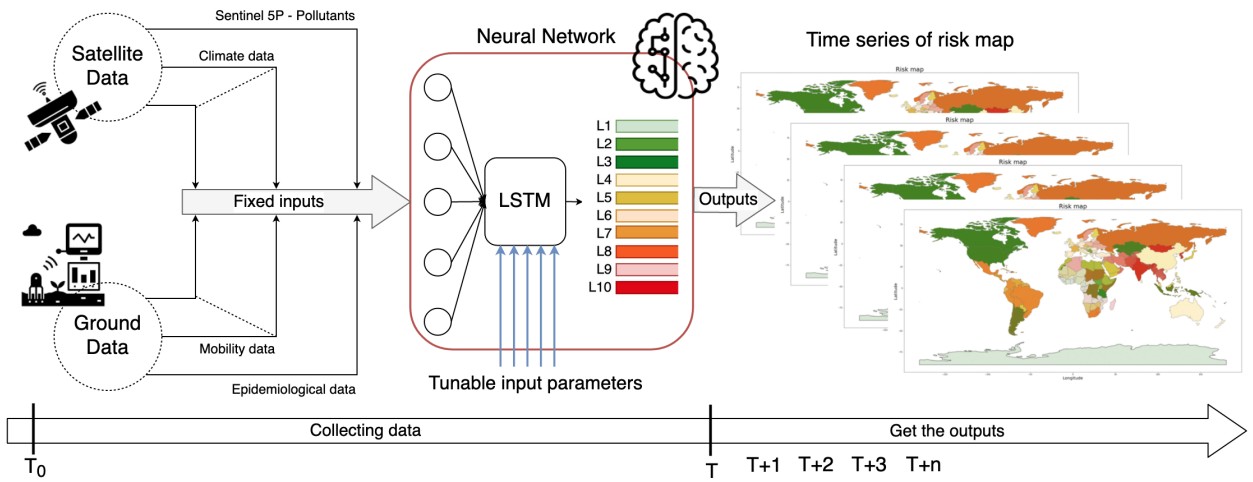

**Figure 2.** Block diagram of the adopted solution.

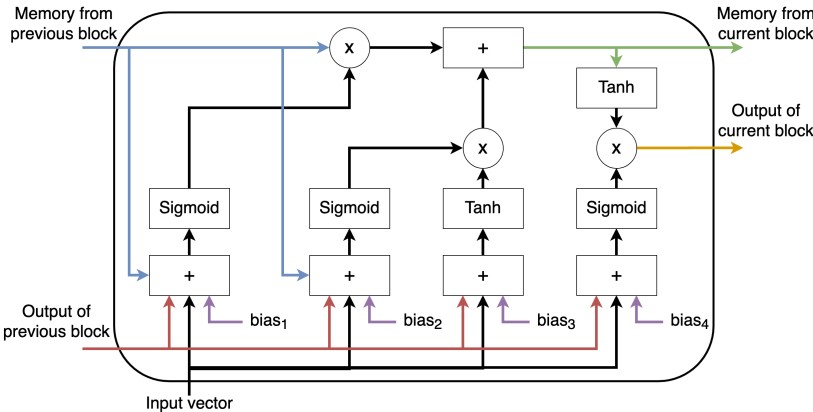

**Figure 3.** Elementary building block of the LSTM network.

## 3. Materials and Methods

The proposed paradigm, as represented in Figure 2, is based on a LSTM network, which was invented to solve the problems of the vanishing and explosion gradients, which the recurrent neural networks (RNNs) suffered [41], and which may allow the model to learn the temporal features from the training data. The choice of this type of network was driven also by the research done on the spread of diseases, as described in several related works [24,27,42,43].

The general architecture shows how several inputs are considered (and their temporal features too), in order to first learn the relationships among the data, and to later produce the risk level as output.

The elementary building block of the LSTM network is shown in Figure 3, where it can be seen that it receives the generic input at the current time step, the output from the previous unit and the memory of the previous unit. Each single building block of the LSTM network makes a decision knowing the current input and the previous output and memory, and it generates a new output and updates its memory.

The functioning of the proposed model is split into two main phases: the training and the prediction. During the training phase, under a supervised learning approach, the model is fed with known output data and the values of the parameters are as specified above. The training set of data depicts the real situation faced in these months, during the COVID-19 pandemic.

Define $X \in R^n$, with $m$ a vector of parameter values, and $Y \in R^p$, with $q$ the vector of outcomes; then the model will be trained to find the correlation between X and Y, as shown by the Equation (1).

$$F \rightarrow Y = F(X) \tag{1}$$

For the model under analysis, we suppose that the input is a matrix $X \in R^{n,m}$, where $n$ represents the number of the different data sources and $m$ represents the size of the acquisitions in the time dimension. The same considerations can be made for the output; indeed, $Y \in R^{p,q}$ where $p$ represents the dimensionality of the output in a fixed instant and $q$ represents the time dimension of the output. In other words, both the input and the output are matrices; the inputs will contain sensed data for a fixed time interval and the output will contain the prediction of the model for a fixed time interval.

The training process can be done with the well known back propagation and gradient descent techniques [44–49]. In a few words, for each training input, the model is updated by using the error and its derivative. The error is calculated through an a-priori-defined loss function. The model will be trained in order to minimize this error. From this last statement, it can be better understood why both inputs and outputs are required in the training phase. For example, given a model $F$ and a set of data (X, Y) with $X \in R^{n,m}$ and

$Y \in R^{p,q}$, during the training, for simplicity, a sample of the dataset is selected. Define $X_i$ and $Y_i$ as the samples; the training phase can be roughly represented by the equations:

$$y_{pred} = F(X_i) \tag{2}$$

$$e = loss\_function(y_{pred}, Y_i) \tag{3}$$

$$\dot{e} = \partial_X loss\_function(y_{pred}, Y_i) \tag{4}$$

Using the error and the derivative of the error, respectively given by Equations (3) and (4), the model weights are updated, until reaching the minimum. After the training phase, the model should be able to generalize, and given new inputs it can generate the new outputs, using the internal state $F$.

The final output of the model will be a time series of risk maps, as shown in the Figure 2, with a resolution that can vary from low resolution intended as risk for a country to high resolution intended as risk for a city, or a smaller area. An example of a single risk map at low resolution is proposed in Figure 4.

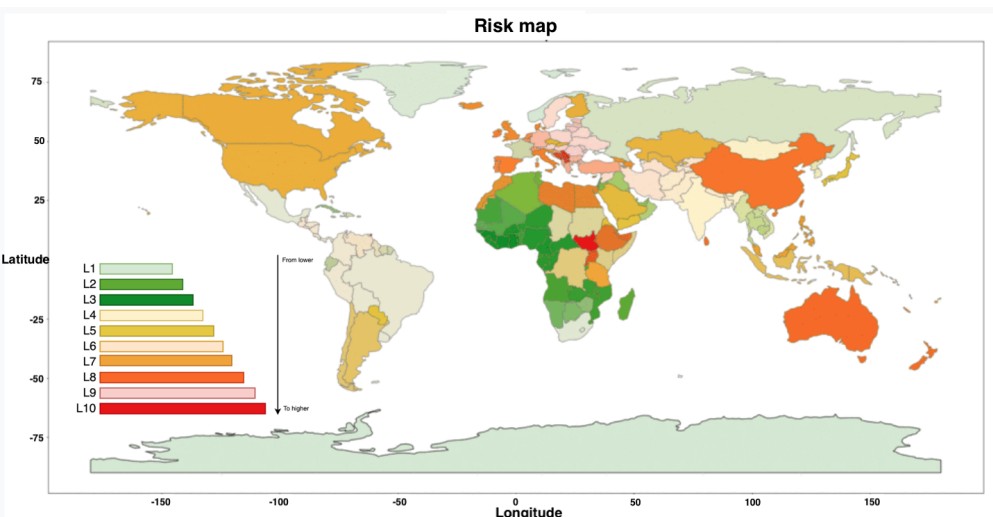

**Figure 4.** Example of risk map at low spatial resolution.

### 3.1. Datasets and Data Pre-Processing

The dataset handling can be split into three main processes:

- Dataset identification;
- Extraction of parameters of interest;
- Data pre-processing.

These three steps are represented in Figure 5, where some data, which will be used for training the model, are presented.

The first step of the processing chain involves the research of datasets suitable for retrieving the data. The examples presented in Figure 5 are related to the mobility dataset (made by Google) [21], the Covid-19 dataset (made by The Johns Hopkins Institute) [22]) and the climate and air quality datasets (stored in the Google Earth Engine catalogs) [16,23]. The second step is mainly dedicated to the extraction of parameters from the datasets. For example, as you can see in Figure 5, climate and pollutant data were extracted respectively from the Era5 and Sentinel-5P Google Earth Engine dataset; the Covid-19 daily new cases were extracted from the Johns Hopkins Dashboard; and the mobility variables were extracted from the Google Mobility Reports dataset. The third step is related to data pre-processing. All the data, extracted before, were processed in the same way in order to have comparable variables. First of all, the data were filtered for the area of interest (AOI) and then only the data related to the time interval of interest were stored. Then, some statistics were extracted from the data, mostly because the data were spatially distributed

(e.g., Sentinel-5P data). For these types of data, the spatial mean, maximum, minimum, standard deviation and median values were computed. Other types of data (e.g., climate data from Era5) can contain multiple acquisitions during a single day; in such cases the temporal means were calculated. In this case, each variable is a vector of daily values, but in general, all the variables are processed in order to fit the lowest temporal resolution from those in the selected datasets. The last pre-procesing step is the temporal averaging. Since it can happen that on some days there are no data, this problem can be overcome by using a temporal average (in the proposed model we chose 5 days), and in addition the not a number (NaN) values are removed.

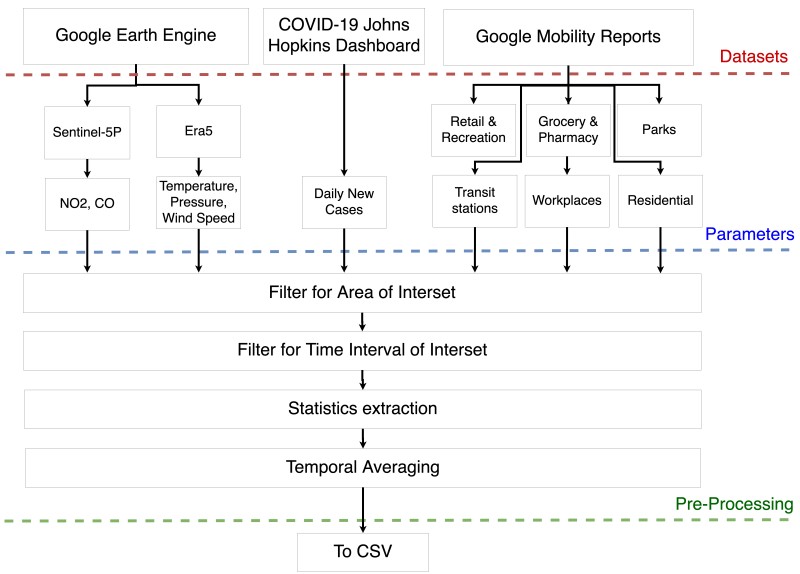

**Figure 5.** Data collection and processing scheme. The overall pipeline can be split into three main steps: source identification, data extraction and data pre-processing.

After these pre-processing steps, the data are structured and stored in CSV files. The structure of the dataset can be seen in Figure 6, showing how the dataset is essentially a 2D matrix, where the rows define the time and the columns the input parameters. The set of parameters is repeated for each AOI.

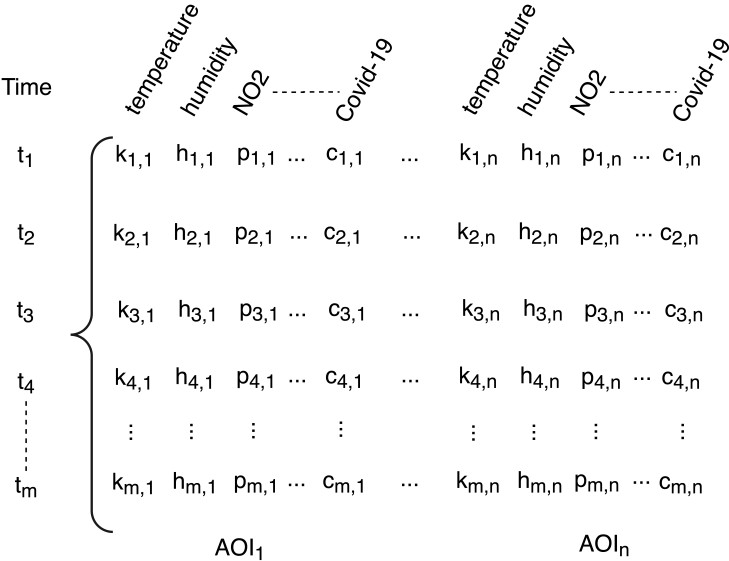

**Figure 6.** Dataset structure.

### 3.2. Training and Forecasting

A first prototype of the model is presented in Figure 7. This model is composed of 3 LSTM layers, with several LSTM blocks, as previously presented in Figure 3; a dense or fully connected network; and a prediction layer. The prediction layer has 10 output nodes, one for each level of risk. The model was trained using the dataset previously described. The inputs, organized as time series, are sent to the LSTM layers, which extract the time information from the data; then by using a fully connected layer, these features are correlated and in the end the prediction layer gives as output the level of risk based on the internal representation of the inputs. For the training process, the model prediction was compared with the ground truth, the true level of risk, defined by an human operator using all the information (e.g., lockdown grade, number of deaths, etc.) concerning the pandemic; then through the back-propagation mechanisms the model weights were updated according to the error between the real level of risk and the predicted one.

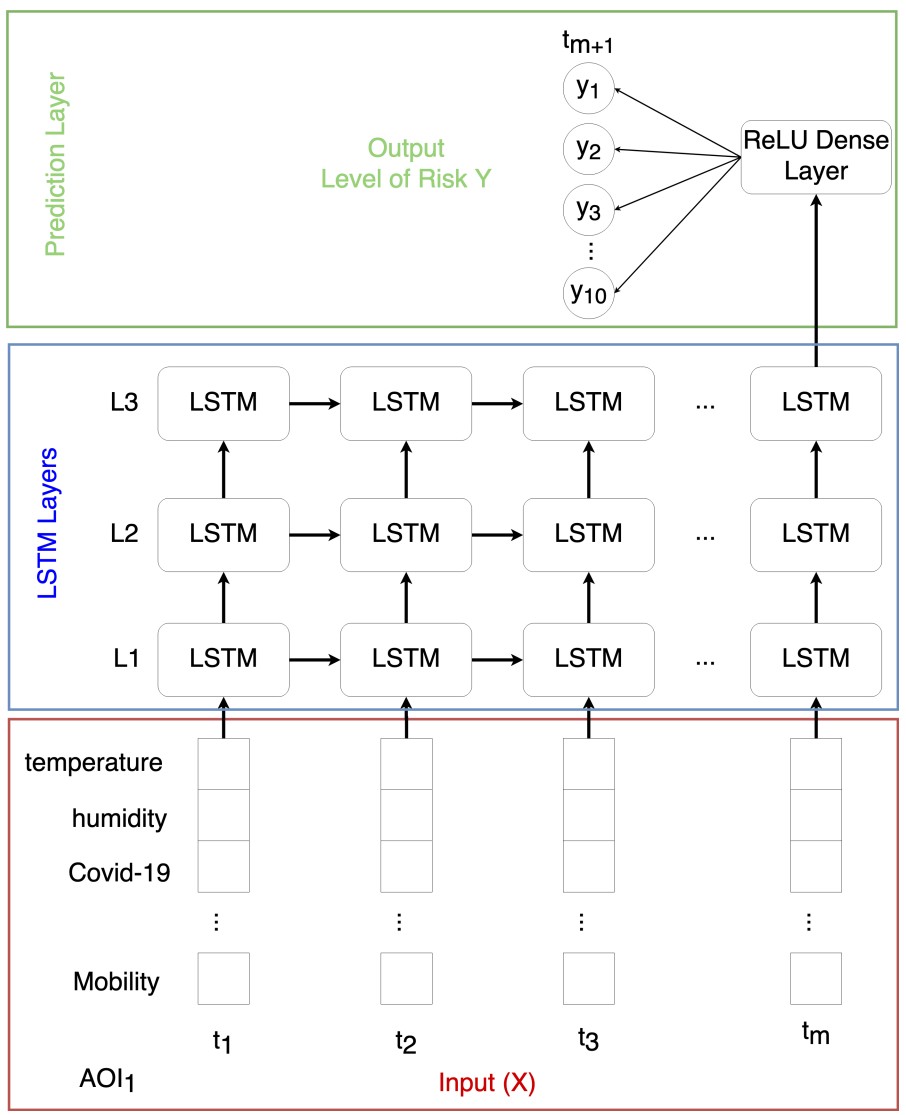

**Figure 7.** Model prototype.

It is important to highlight that after the training process, the model can be used to predict the level of risk of a region never used for the training phase. This is one of the most important aspect when using AI. Given the new inputs from time $t_1$ and time $t_m$ of a new region, the model is able to predict the level of risk in a future time interval.

The main advantages of the proposed model and method are:

- LSTM networks, like RNNs, can learn temporal features;
- After the training phases, which can take a long time (e.g., some days), the prediction phase is more rapid (fraction of second);
- The LSTM overcomes RNN faults;
- The model is highly scalable and can be used in different scenarios.

There are also some disadvantages in using AI and LSTM:

- LSTM has some limitations in capturing the data relationships between different data series when the delay is too big, so it could be necessary to introduce attention layers;
- The AI model produces output that cannot be explained—for that reason, the so called "explainable AI models" are under development;
- A partial re-training of the network may be necessary if applied to a very different context than the one on which it was trained.

Since every nation, region or city differs from every other and its boundary conditions will certainly be different, it may be necessary to re-train the network, even if only partially. In fact, in this case it is possible to reuse a previously trained network to specialize it for a particular regime of use, through the well-known technique of transfer learning [50,51]. This technique allows one to use the weights of the previously trained network, to lock some layers and modify or add others, in order to adapt the network to the particular scenario at hand. This means that the training process for a new scenario is much faster than the total training.

Even if applied to image analysis, there are several works on transfer learning for the recognition of some features in COVID-19 patients [28,52,53], which made us think to positively apply this technique to our model.

## 4. Preliminary Analysis

As previously discussed, our proposal was born in response to a MIUR call on COVID-19 and aims to create a multiscale analysis system based on AI algorithms, to provide an a priori tool for decision makers. The needed information is based on the combination of multi-source data with measurements obtained from fixed and mobile networked sensors, from satellites and through proximity surveys, when necessary. Such a proposal required the integration of multi-disciplinary skills, ranging from the development of sensor networks and satellite data processing to particle detectors; biochemical analysis of particulate matter; innovative strategies in the environmental field; and sophisticated computational technologies for treatment, analysis, and interpretation of data by using big data technologies.

We want to highlight the complementarity of our workgroup, made up of three research teams, which cover satellite monitoring and the development of data fusion and crowd monitoring models (University of Sannio), the miniaturization and networking of high-resolution sensors on board drones (Politecnico of Milano) and biochemical, health and environmental issues (University of Bologna). This high level of interdisciplinarity will help the project to succeed. The idea behind this work was to submit a "Concept Paper" with the aim of finding interested people to collaborate with in the further research and development of the project, if financed. Much of the work must still be done, at least in reference to the realization of the neural network and to the creation of all necessary datasets. However, some progress has been already made, and in the next paragraphs the state of the work will be presented.

### 4.1. Measuring Pollutants, Air Conditions and Pandemic Data

Since the core of the idea involves the use of pollutant measurements and the infection data, beyond the involvement of other information, as already highlighted, the authors first selected a list of possible pollutants, such as nitrogen dioxide ($NO_2$), sulfur dioxide ($SO_2$), formaldehyde (HCHO) and ozone (O3), which can be retrieved through the use of Sentinel-5P, and other data (for instance, PM2.5, PM10, etc.) which can be retrieved by using other sources, at different multi-scale levels.

The purpose of the measurement of pollutants is twofold. As it will be discussed ahead, some pollutants favor the transmission of the respiratory diseases, and therefore by measuring them and feeding the model with their values, a contribution to the final risk level is obtained. Practically, we analyzed sets of parameters to select those which were more efficient in training the neural network, since they already present clear inter-relationships.

In addition, some pollutants give us information on the degree of lockdown. Again, the model is able to trace the goodness of the lockdown by using as input certain types of pollutants, and to capture the inter-relationships among the data. What is important is to understand that the analysis wants to highlight two different levels of vision: macro high-level vision (i.e., through the use of satellites, for instance, or data at a wide geographical level, national, etc.), and micro low-level vision where local understanding of the phenomenon is carried out. The proposed tool, based on AI, and the specific neural network, can be used at each level, once trained, and give important insights. In this regard, an interesting analysis has been done in [54], which discusses how ML algorithms can help in characterizing airborne particulates for applications related to remote sensing.

In our case, we started analyzing the trends of some pollutants in specific regions of interest. Examples of such data, retrieved both through ground and satellite platforms, are shown in Figures 8 and 9 respectively. Figure 8 shows an air quality map at the European level, retrieved through the European Environmental Agency website [55], and Figure 9 presents the levels of some pollutants obtained processing the Sentinel-5P satellite data.

Regarding the COVID-19, we made use of the public Johns Hopkins Dashboard, shown in Figure 10, which allows one to download the contagion data organized per state, region and city, and contains much more information, such as the daily new cases, daily new deaths and so on, in all the world [22].

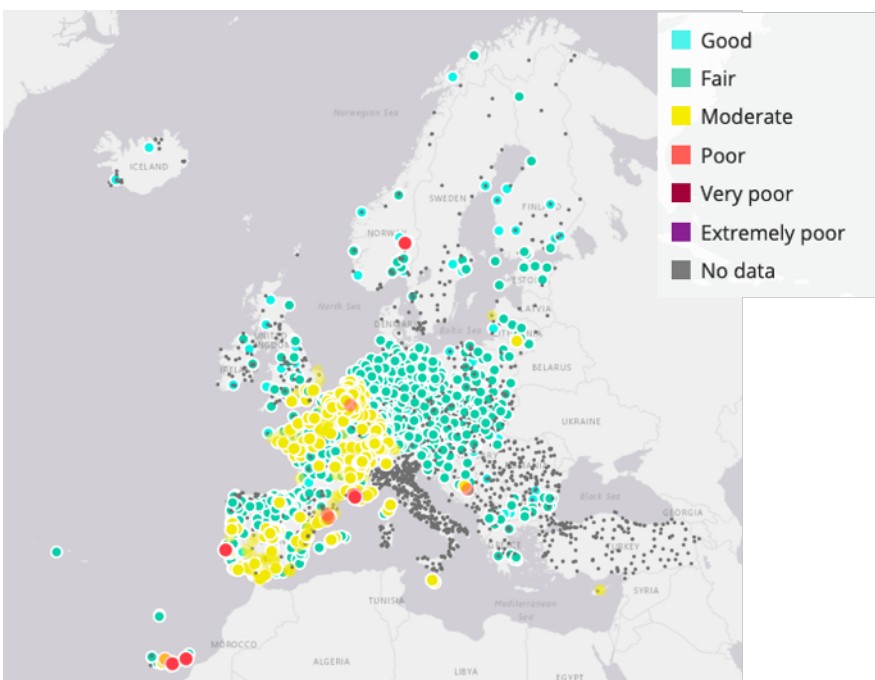

**Figure 8.** Air quality map from the European Environment Agency website [55].

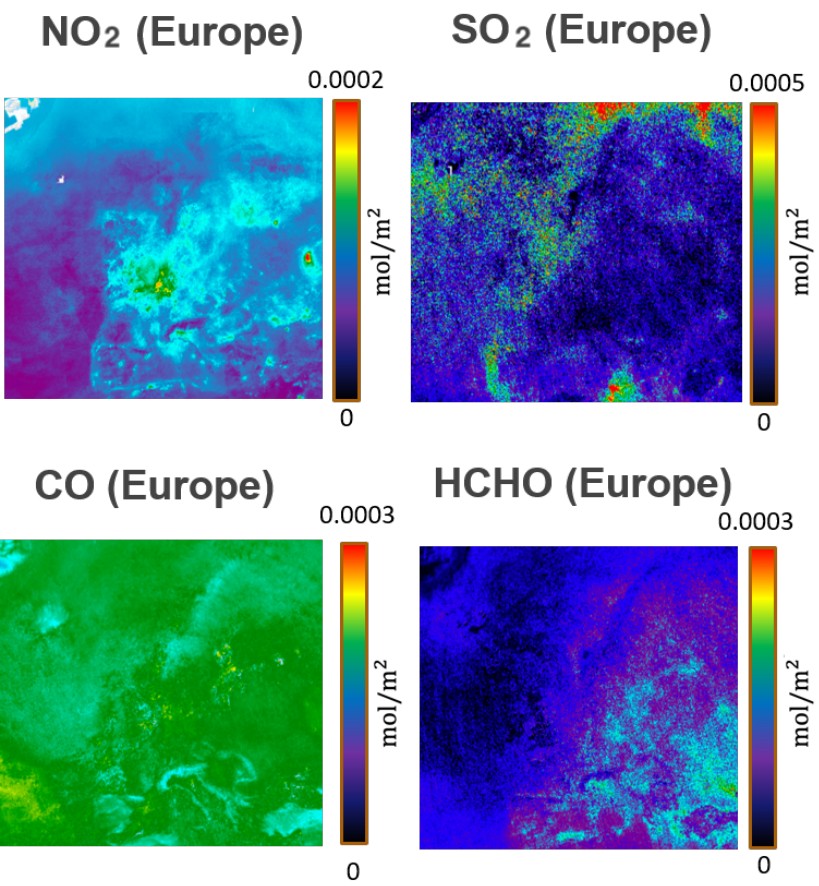

**Figure 9.** Sample of Sentinel-5P pollution data.

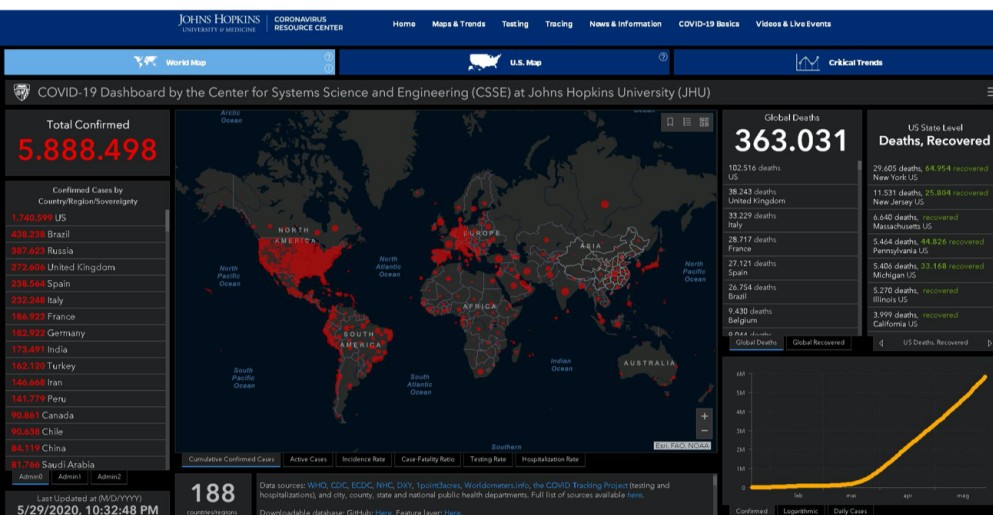

**Figure 10.** COVID-19 Johns Hopkins Dashboard [22].

### 4.2. Influence of Air Quality on COVID-19 Diffusion

About the relationship between pollution and COVID-19 diffusion, there are several epidemiological studies dealing with the different components of the atmosphere, and their correlations with the spread of this disease. The most important indicators of air pollution, which showed significant positive correlations, are fine particulate, nitrogen oxides, carbon monoxide, ozone and volatile organic compounds [32,56,57].

An accurate study, considering strengths and limitations of similar statistical analysis, taking into account many confounding parameters, showed in particular a strong association between PM2.5 exposure and mortality rate [58].

Furthermore, other environmental and meteorological conditions have been considered. A diffuse consensus about a positive correlation between wind and COVID-19 infection, and a negative correlation between solar irradiation and the disease is present, and the roles of humidity, temperature and rain are still debated [57].

At the moment, it can be concluded that many of these hypotheses could be valid, and therefore various factors could be responsible for the spread of the infection. In any case, we considered it very important to focus on the NOx and fine particulate presence in the air. For this reason, some work has been done to analyze a possible correlation between the virus and the concentrations of NOx and particulate matter.

### 4.3. Particulate Matter (PM)–Virus Correlation

More than 240 scientists recently signed a Commentary, appealing to the medical community and highlighting the need for dealing with the airborne transmission of SARS-CoV-2 [59]. Many researches have proven that the previously indicated "safe distance" of 6 feet cannot be considered sufficient, since different ways of diffusion can occur in indoor and outdoor environments [60]. Since the beginning of the COVID-19 pandemic, several studies have been carried out to investigate the reasons for the uneven distribution of infections and fatalities within the different countries, and positive correlations with air pollution, particularly with suspended fine particles, have been found [61–65]. Some researchers explain these results considering acute and chronic effects towards the respiratory system that could make it more susceptible to pathogen infection, while others suggest that different biotic and abiotic factors could be inhaled adhering to the already suspended fine particles [66].

Even though the modes of COVID-19 transmission are still under discussion [67–69], the possibility of considering the presence of SARS-CoV-2 RNA on PM10 in outdoor environments has also been suggested [70]. Many previous studies already demonstrated that different varieties of microorganisms are present on the suspended particulate [71], and recently early warnings were given about the possibility that they could play a role of carrier for the coronavirus [72], supported by the finding of SARS-CoV-2 or viral nucleic acid on particulate samples in outdoor environments [73,74].

According to the cited researches, both PM10 and PM2.5 could be relevant to improving viral infectivity. However, it has been hypothesized that the effect is not linear in all conditions, but that the prolonged high concentration of fine particulate (for instance, more than the daily limit of 50 $\mu g/m^3$ of PM10) could trigger a "boost" effect on the spread of virus [62].

## 5. Case Studies: Correlation between NO$_2$ and COVID-19 Data in Two Regions of Interest

As highlighted before, it is extremely important to change the level of analysis, from a macro to a micro-level, to better understand and monitor the evolution of the phenomenon, and to get the right input values for the DSS in such a way that the initial diffusion of the virus can be detected and stopped with targeted levels of lockdown or the adoption of other specific measures.

### 5.1. Macro-Analysis Based on Satellite Data

In the first part of our analysis, a study was carried out to evaluate the correlation between one of the pollutant, the NO$_2$, and COVID-19. This preliminary analysis must not be surprising. In fact, even if it is true that the neural network has the task to find the inter-relationships among the data, a pre-selection of these data work to improve the training phase of the neural network.

Sentinel-5P satellite data related to the NO$_2$ concentrations have been acquired, processed, averaged and plotted by using Python scripts, through the Google Earth Engine

(GEE). In Figures 11 and 12, the concentration of NO$_2$ is shown for Italy and China respectively, over two similar periods. It is possible to see that in the worst period of pandemic for both countries, the highest concentration of NO$_2$ laid in the regions where COVID-19 had its highest values (Lombardy region and Wuhan).

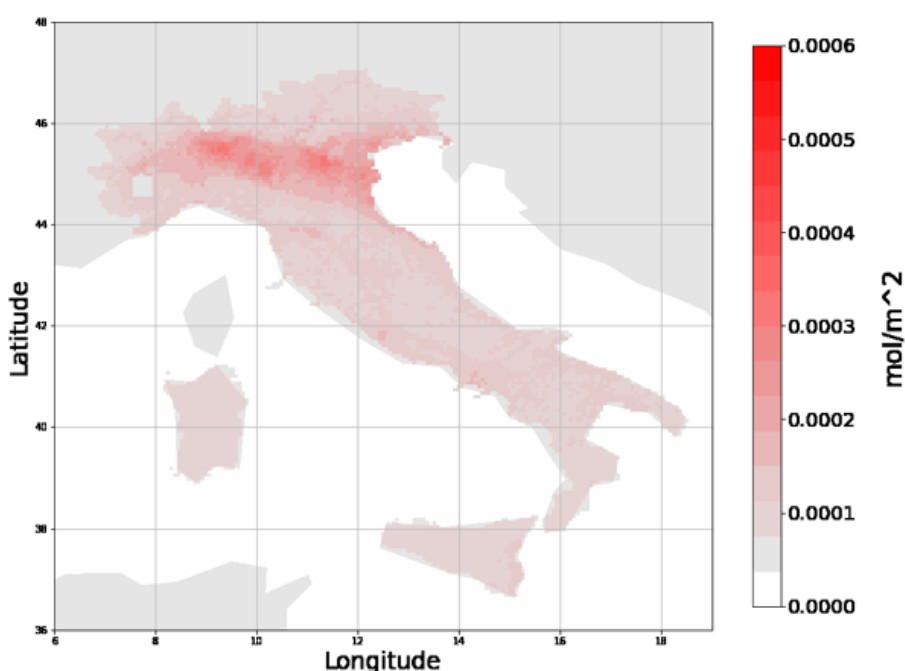

**Figure 11.** NO$_2$ concentration in Italy (from 1 to 5 January 2020).

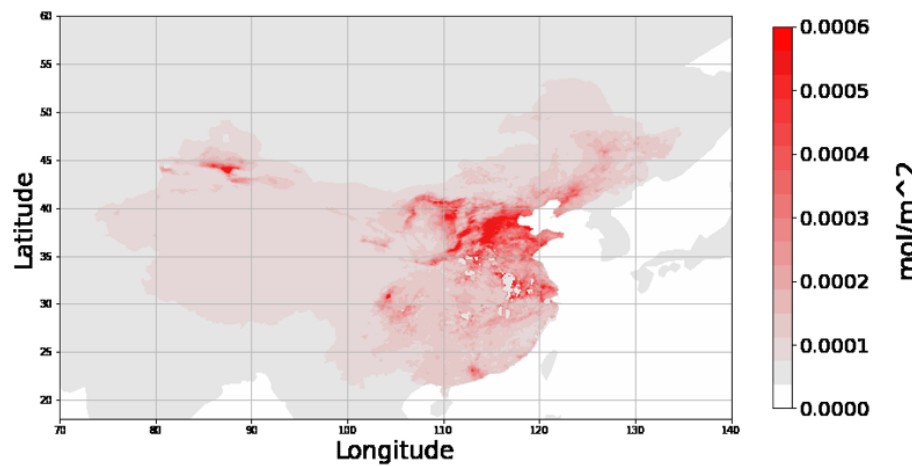

**Figure 12.** NO$_2$ concentration in Hubei, China (from 1 to 8 January 2020).

For this reason, rather than analyzing the data relating to the entire nations, we have chosen to focus on the regions in Italy and China where the spread of the virus has been most rapid and emblematic in the first months of the year.

Since Sentinel-5P covers wide areas, a subset has been created to obtain the NO$_2$ concentrations related to the chosen regions. Some statistics on the subset have been calculated, including the maximum, the minimum and the standard deviation. The maximum

values of the pollutant have been plotted in Figures 13 and 14 for further discussion. In the two figures, to manage the absence of some values in Sentinel-5P data, the average concentrations for the $NO_2$ over five days have been calculated, by starting from 1 January and continuing until 30 June 2020.

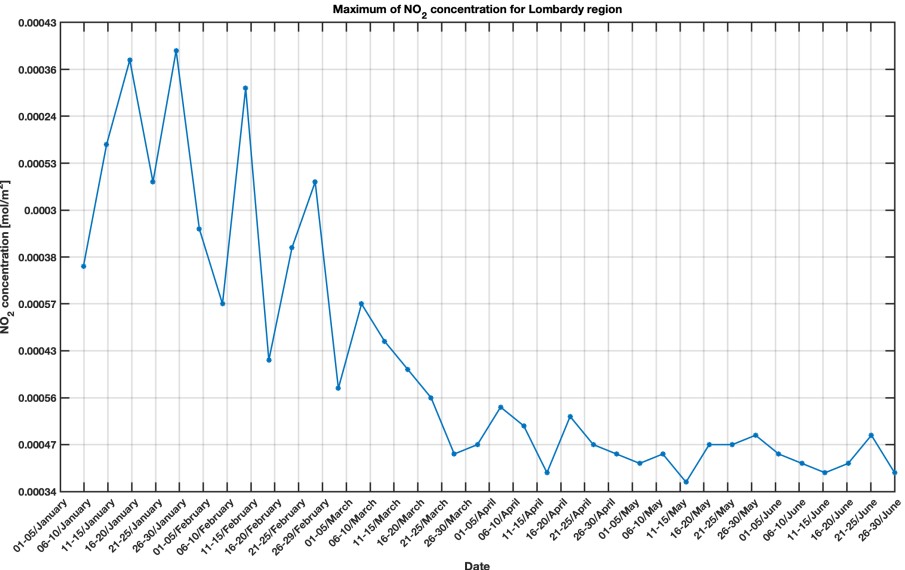

**Figure 13.** Maximum $NO_2$ concentration for Lombardy region.

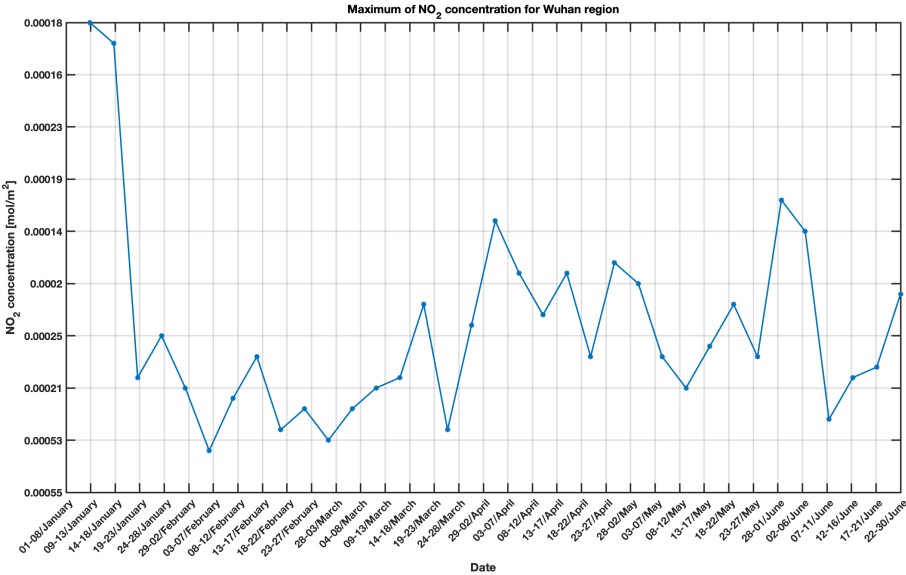

**Figure 14.** Maximum $NO_2$ concentration for Wuhan region.

The correlation between the peaks of the average $NO_2$ concentration and the number of new people testing positive to COVID-19 has been analyzed. To match the satellite and epidemiological data, it was necessary to make an average over 5 days also for the number of new infections. Several scatter plots have been constructed using these data, positioning the number of new infected people on the abscissa axis and the $NO_2$ concentration on the ordinates.

Moreover, to take into account the delay between the instant when infection occurs and the actual evidence of the transmitted contagion, a delay analysis has been carried out: the COVID-19 data were moved forward in time, thereby creating scatter plots with different delays.

In Figures 15 and 16, the scatter plots each present a sequence of data, where the delay unit (corresponding to 5 days) is defined by the number near the dot. In the first Figure an offset of nine delay units has been applied between the two series of data, while an offset of seven delay units has been applied in the second case. This representation helps to graphically catch the correlation between $NO_2$ and COVID-19 and also to identify emblematic cases like the Wuhan ones. In fact, in this latter case the trend, at a certain moment, starts oscillating.

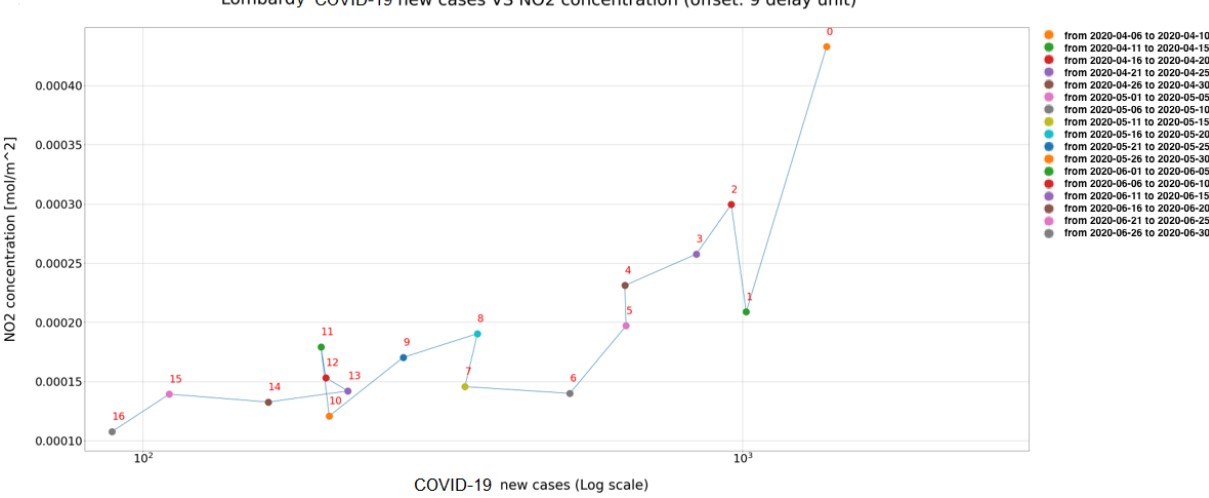

**Figure 15.** New daily cases vs. average concentration of $NO_2$ in Lombardy.

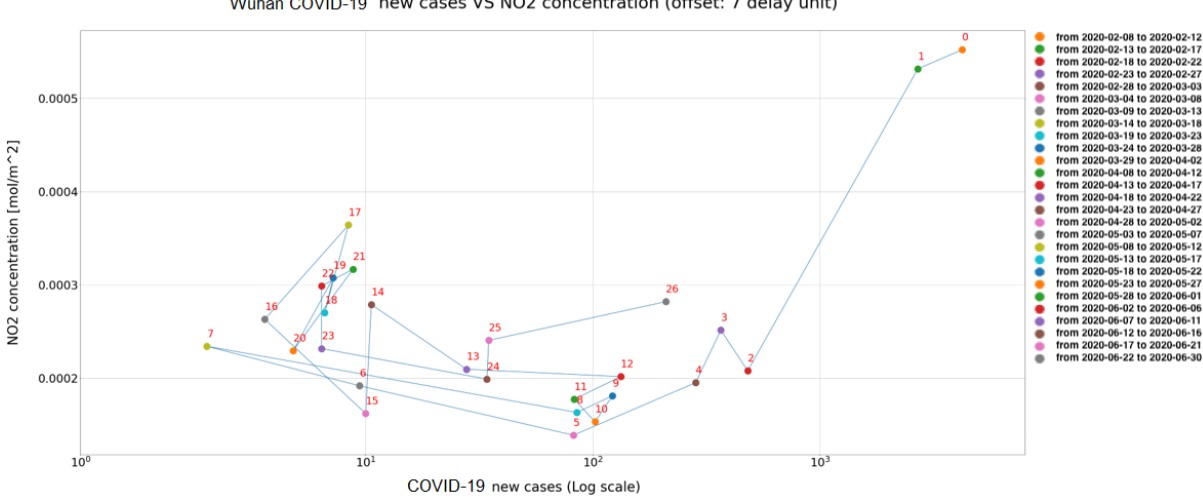

**Figure 16.** New daily cases vs. average concentration of $NO_2$ in Wuhan.

This phenomenon can be better explained by analyzing Figures 17 and 18, where it is evident that while for the Lombardy region both the $NO_2$ and COVID-19 have a decreasing trend over time (from a high number of infected and a high concentration of pollutant to a low number of infected and a low concentration of pollutant), for the Wuhan region the situation is quite different and unexpected, since after a certain period the $NO_2$ starts increasing again while the new infections are stable and close to a very low value.

Several interpretations can be given for this latter result: (1) at Wuhan the adopted countermeasures were able to contain the virus, and were sufficient to release the lockdown; (2) people knew how to comply with the rules (e.g., wear the mask) avoiding the rise of the number of infections; (3) the COVID-19 data communicated in the second period were not correct. Actually, we cannot decide which hypothesis is the right one. In our opinion, the first two are the most likely.

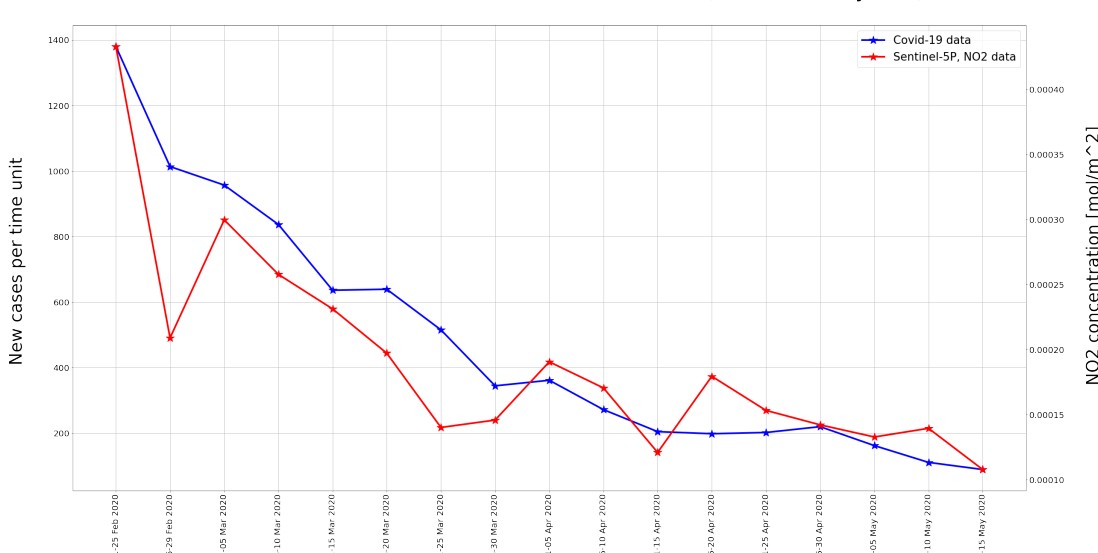

**Figure 17.** New daily cases vs. average concentration of NO$_2$ in Lombardy.

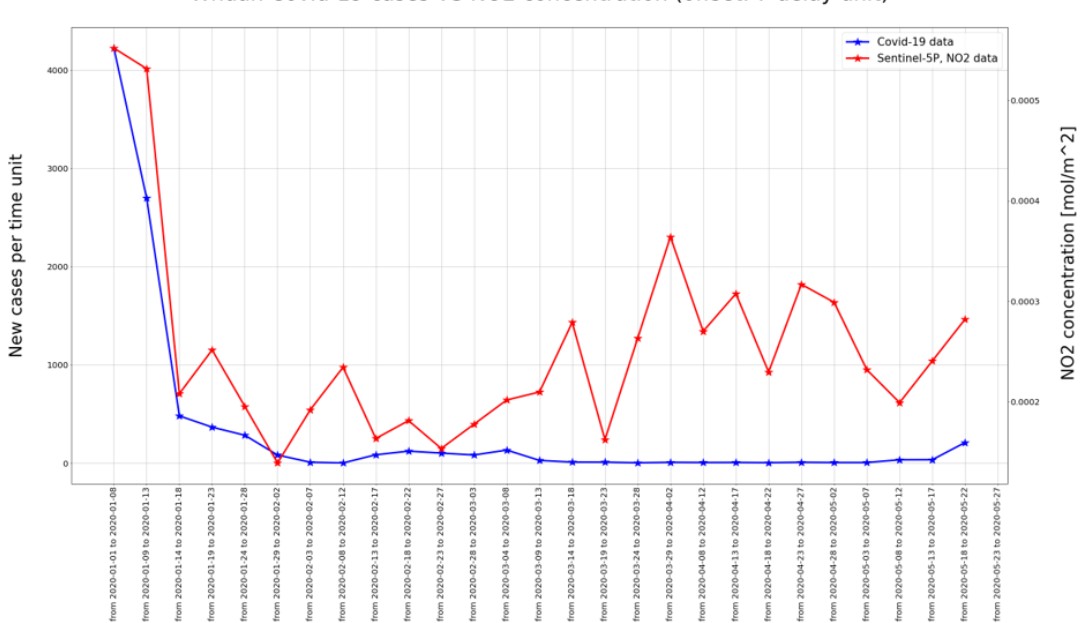

**Figure 18.** New daily cases vs. average concentration of NO$_2$ in Wuhan.

The offset values, used to design the scatter plots of Figures 15 and 16, were decided through the analysis of the correlation between the new COVID-19 cases and the NO$_2$ concentration. This correlation was calculated by using the Pearson correlation coefficient (PCC), and as specified before, applying a delay analysis.

From Table 1, it can be seen that the maximum correlation (positive) value was recorded with a delay of nine units in the case of Lombardy region, and with a delay of seven delay units in the case of Wuhan.

Further considerations can be made. First of all, if the delay units are converted back to number of days, the maximum value of the correlation between NO$_2$ and COVID-19 is found after 45 days for the Lombardy region, and after 35 days for the Wuhan region. Somehow, it seems that the adopted restrictions in Wuhan brought about better results in terms of COVID-19 reduction than in Italy. Moreover, it is clear that the a posteriori measures of lockdown cannot be an efficient means of intervention, because before the

effects of the infection rate become significant to the observers, after more than one month, one month and half must pass. This latter consideration works in convincing us that a posteriori measures are belated and cannot be considered efficient.

**Table 1.** Pearson correlation coefficient for Lombardia and Wuhan (maximum correlation in bold).

| Delay Unit | PCC Lombardia | PCC Wuhan |
|:---:|:---:|:---:|
| 0 | 0.0770 | −0.2474 |
| 1 | 0.2773 | −0.2514 |
| 2 | 0.4983 | −0.2496 |
| 3 | 0.6629 | −0.1775 |
| 4 | 0.7180 | 0.0271 |
| 5 | 0.7918 | 0.1934 |
| 6 | 0.8774 | 0.3842 |
| 7 | 0.8092 | **0.4857** |
| 8 | 0.7532 | 0.3905 |
| 9 | **0.8969** | 0.2969 |
| 10 | 0.8334 | 0.2905 |
| 11 | 0.8403 | 0.1813 |
| 12 | 0.8736 | −0.0652 |
| 13 | 0.7830 | −0.2000 |
| 14 | 0.8302 | 0.0291 |
| 15 | 0.8854 | 0.1762 |

*5.2. Micro-Analysis Based on Networked Sensors*

Given all the above considerations, and by taking into account that not all the pollutants can be recovered through a satellite analysis, and that different levels of investigations may become necessary to target the measure at the local level, it is important to discuss the characteristics that the sensor networks must hold in order to guarantee the collection of the pollutant values in a way that is useful for the intended purposes.

In regard to the sensors used at ground level, the main challenge, and a significant element of innovation, concerns the identification of dust sensors, in particular for PM2.5, of a limited cost, power consumption and footprint for a widespread installation in both fixed and mobile (for example with drones) wireless networks, to investigate geographical areas of particular interest, while maintaining peculiar characteristics such as selectivity and sensitivity in the measurement.

Traditionally, PM is measured by means of the laser scattering technique, and particles are hydrodynamically concentrated to flow in a single stream on which the laser beam is focused. The presence and size of each single particle can be determined from the intensity of the scattered light pulse, assuming a spherical shape and average optical properties. The granulometric distribution is thus obtained with reasonable accuracy, and the smallest detectable diameter is diffraction-limited, in the order of the light wavelength, i.e., hundreds of nanometers. The laser scattering technique represents the state of the art for particles in the 0.3–10 μm size range. However, due to their cost and bulkiness these instruments are not suitable for massive deployment in the environment. They are typically installed in a few fixed monitoring stations controlled by local environmental protection agencies.

Given the relevance of air pollution for human health and the need for better spatiotemporal resolution in mapping complex phenomena, such as dust generation, concentration and transport, several efforts were carried out in the last decade to develop compact and affordable devices for measuring PM in a distributed, pervasive way. Recent implementations of wireless sensors networks at the city scale have demonstrated the feasibility of this paradigm [75–80], and specific technologies have been leveraged to address several challenges, in particular to preserve the environment and human health, spanning from monitoring air, water [81] and the surrounding environment to controlling natural disasters and preserving landscape resources [82].

The employment of miniaturized devices has spread out, with different characteristics, mainly grouped into two classes. Micro-machined silicon-based sensors represent the ultimate degree of miniaturization and leverage microfabrication capabilities to enhance the detection sensitivity [83]. Two main approaches have been proposed for solid-state detection: the use of mechanical resonance in oscillating micro and nano-weighting scales and high-resolution capacitance measurements of the single-particle impedance on chip [84]. Despite very promising preliminary results and potential for nanoparticle detection and ubiquitous integration in handheld devices, such as smartphones, thanks to their millimetric size, they still appear far from commercial maturity. A critical aspect, in addition to clogging, cleaning, lifetime and power dissipation, when shrinking down the sensor size, remains the active fluidics necessary to capture and collect dust particles in the chip.

Another more consolidated class of PM sensors is that of low-cost optical sensors, named optical particle counter (OPC). They represent an evolution of smoke detectors in which a photodetector measures the amount of scattered light from the dust when illuminated by a LED. Several sensors of this class are available on the market from companies such as Alphasense, Honeywell, Plantower, Sharp and Shinyei. The latter are very compact (credit-card sized, Figure 19a,b) and consolidated, enabling new versatile scenarios (Figure 19c) in pervasive monitoring of dust concentration, also capable of coping with emergency situations such as the acute phases of pandemics.

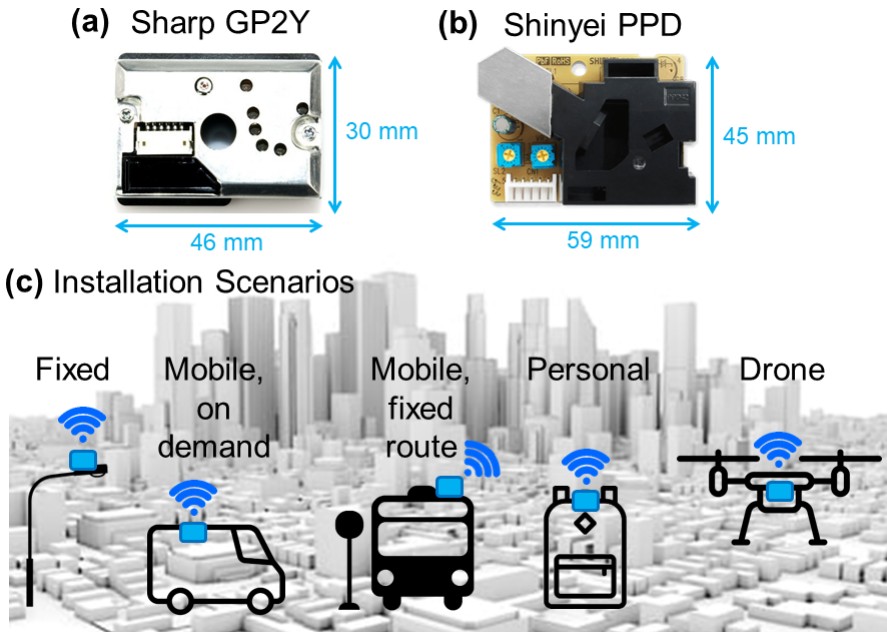

**Figure 19.** Examples of low-cost, credit-card sized PM sensors from Sharp (**a**) and Shinyei (**b**) enabling new versatile scenarios (**c**) in the pervasive monitoring of dust concentration, which are also capable of coping with emergency situations such as the acute phases of pandemics.

Different works have compared their performances with reference instrumentation, both in the laboratory [85,86] and in the field [87], finding good agreement. In the majority of conditions, they can quantify the concentration of particles with a size range similar to that of laser scattering, with a full scale of about 1000 $\mu g/cm^3$, and a sensitivity of a few tens of $\mu g/cm^3$, well matched with the regulatory limit of 50 $\mu g/cm^3$. Furthermore, the response time is in the order of seconds, fast enough to capture the dynamics of human and air transport. Thus, they are suitable for the purpose of this project, enabling the deployment of thousands of sensing nodes capable of quickly detecting exceeding of PM10 and PM2.5 concentration limits. Indeed, artificial intelligence can leverage the high level of redundancy and overlap in spatial mapping to compensate for the intrinsic limits of this class of devices.

### 5.3. Correlation between Mobility and COVID-19 Data

Finally, a further experiment was conducted. In this case, the aim was to search for a possible correlation between the trend of new COVID-19 positives and mobility data. As discussed in Section 3.1, mobility data were acquired by processing the Google Mobility Report [21]. This dataset contains a worldwide mobility report since the COVID-19 outbreak. Each report consists of a time-series of six variables indicating mobility; they are: retail and recreation, grocery and pharmacy, parks, transit stations, workplaces and residential. Each variable is expressed as a percentage change from the baseline, calculated in the previous years.

In Figure 20, the trends of mobility data for the six variables above specified are shown for the Lombardy region, over the same period presented in Figure 5.

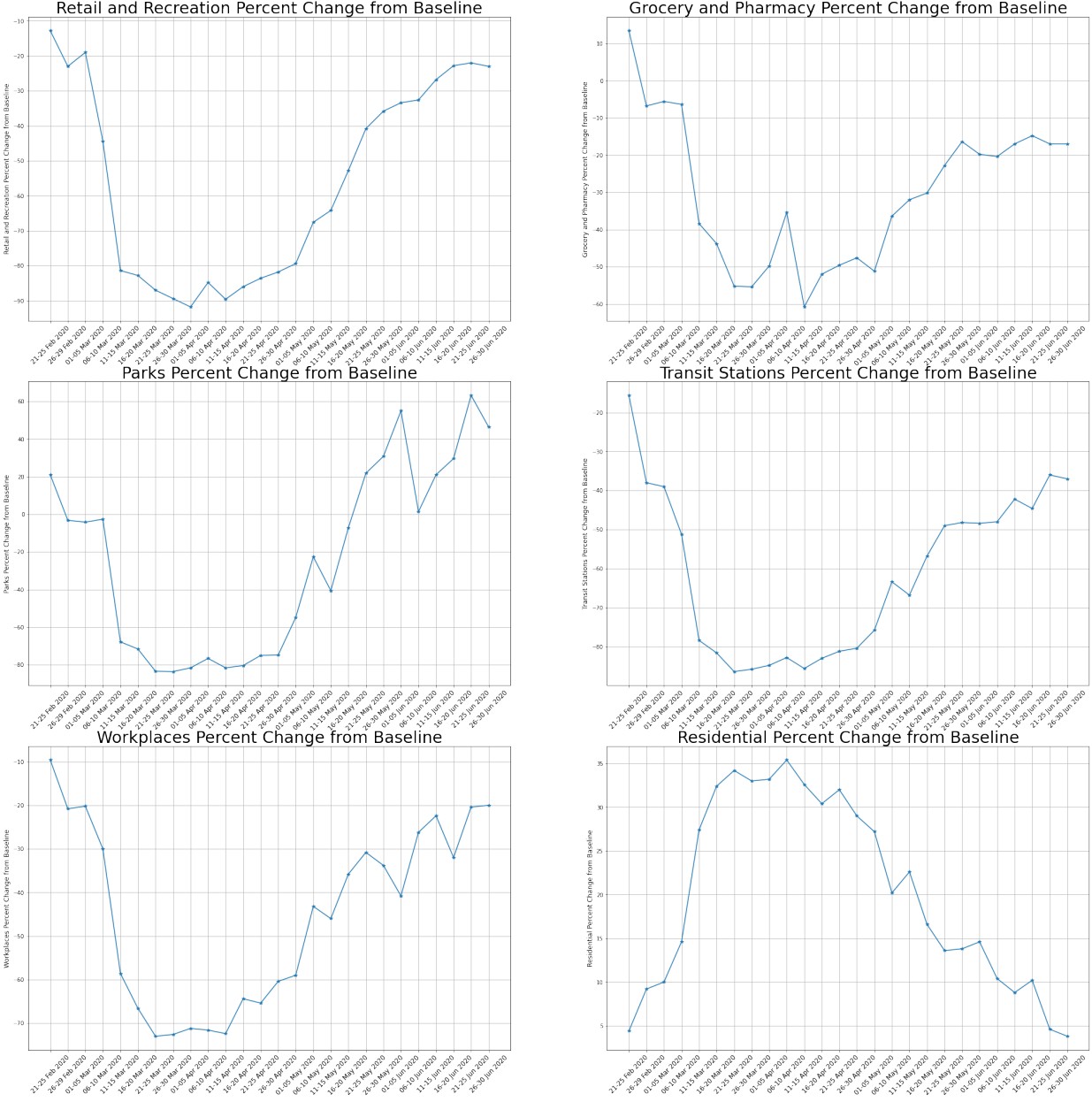

**Figure 20.** Mobility data for the Lombardy region (from 20 February 2020 to 30 June 2020).

Following the same principle of Section 5.1, in this case a scatter plot with the delay analysis was computed. In this case study the maximum of correlation was found after

12 delay units (60 days), as can be seen in Table 2. The plot corresponding to the maximum correlation between mobility data and COVID-19 data is shown in Figure 21.

**Table 2.** Pearson correlation coefficient for Lombardy.

| Delay | PCC | Delay | PCC | Delay | PCC | Delay | PCC |
|---|---|---|---|---|---|---|---|
| 0 | −0.7516 | 4 | 0.2514 | 8 | 0.6045 | 12 | 0.9186 |
| 1 | −0.5240 | 5 | 0.3773 | 9 | 0.7451 | 13 | 0.9045 |
| 2 | −0.2078 | 6 | 0.4358 | 10 | 0.7879 | 14 | 0.9008 |
| 3 | 0.0654 | 7 | 0.4805 | 11 | 0.8837 | 15 | 0.9051 |

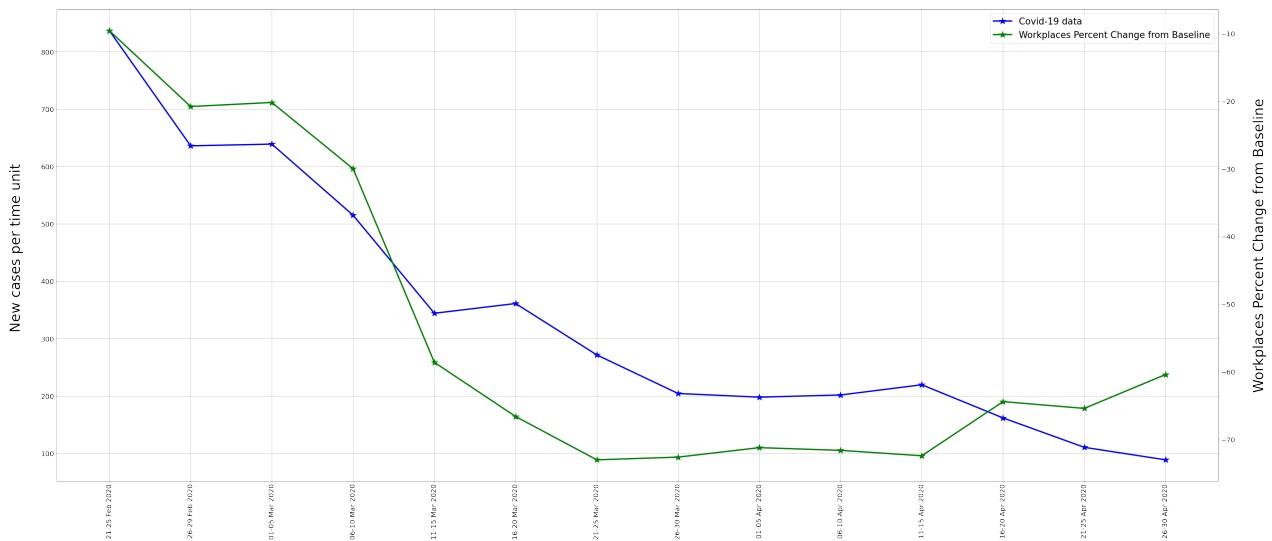

**Figure 21.** Workplaces vs. daily new cases in Lombardy.

## 6. Discussion

Some additional considerations are given in this section, although many relevant points have been already highlighted in the previous section where preliminary results have been presented.

As analyzed in detail, the proposed approach aims to develop a novel model for the cooperative fusion of extremely heterogeneous data, both in terms of nature and source (epidemiological data, environmental data and data related to the human activities), and in terms of spatial (km to m) and temporal (days to seconds) sampling, to capture the extremely complex dynamics, difficult to identify with other traditional methodological tools. While we could find some examples of similar works aiming to realize a DSS to benefit decision makers, none of them are based on AI algorithms able to work on huge amounts of data and to use the necessary granularity based on time unit of 24 h or less. In fact, the algorithms of AI are able to extract the features related to the hidden correlations between several elements and different data, such as the concentrations of atmospheric particulate matter, the meteorological trends and the virus spreading, to estimate the level of risk. Moreover, many new satellites recently launched into space have very low revisit times by offering swift responses in terms of available data and necessary information to extract.

Besides that, it has been explained that the designed model is expected to operate on two levels of analysis:

- Macroanalysis: mainly through satellites (not limited to) with the data collection over wide areas.

- Microanalysis: through the use of fixed or dynamic networks for local-focus data collection.

Therefore, the final output of the model will be a time series of risk maps, with a resolution that can vary from low resolution intended as risk for a country to high resolution intended as risk for a city, or a smaller area. The final resolution will be linked to the sensor with the worst resolution. Then the data will be processed to match that resolution. To obtain an overall picture, both nationally and globally, the individual predictions made at this resolution can be mediated. Obviously, since this is a concept paper, this solution must be further investigated and it may be necessary to train different models at different resolutions.

In fact, the two case studies presented in the above section referring to Lombardy and Hubei regions respectively, aimed to make some comparison in terms of lockdown measures and impacts on COVID-19 reduction. However, it is worth underlining that the idea was to initially develop the tool with particular attention to the most critical areas of Italy, but later it could also be used in different parts of the world once the neural network has been properly trained on the different conditions and its parameter set properly.

The final step of this work, as anticipated in the Abstract, will be the implementation of a cloud-based system. This system will give the possibility for a user to log-in to a personal work-space and to perform the analysis. The platform will have a practical and easy to use GUI (graphical user interface) that will help users in doing their analysis. The core of the system is the neural network model, but the GUI will help the user to interact with it, by selecting and tuning special inputs in the easiest way.

## 7. Conclusions

In this manuscript, we have presented the cross-disciplinary AIRSENSE-TO-ACT project, aiming at the creation of a decision support system for the timely and effective activation of targeted countermeasures during viral pandemics, such the COVID-19, based on a model merging data from very heterogeneous sources, including ground wireless networks of low-cost dust monitors, and satellite data, spanning from meteorological and pollution data to crowd sensing. The correlation between virus diffusion and concentration of $NO_2$ has been analyzed, and the further analysis of the correlation between virus diffusion and PM in the air would be the main focus of future investigations. However, the proposed model goes well beyond that, since it aims to collect as inputs information such as mobility, weather conditions, air quality, infection numbers, etc., in order to capture all the possible interactions and propose the level of risk as output, based on a micro and macroanalysis for targeted measures. The "second wave" currently spreading in Europe demonstrates the urgent need for a tool such as the one proposed in this paper, in order to limit the economic damage of generalized lockdowns and restrictions, and above all to avoid huge numbers of dead people.

**Author Contributions:** Conceptualization, Fabrizio Passarini, Marco Carminati and Silvia Liberata Ullo; Data curation, Alessandro Sebastianelli, Francesco Mauro, Gianluca Di Cosmo and Silvia Liberata Ullo; Formal analysis, Fabrizio Passarini, Marco Carminati and Silvia Liberata Ullo; Funding acquisition, Silvia Liberata Ullo; Investigation, Fabrizio Passarini, Marco Carminati and Silvia Liberata Ullo; Methodology, Alessandro Sebastianelli, Fabrizio Passarini, Marco Carminati and Silvia Liberata Ullo; Project administration, Silvia Liberata Ullo; Software, Alessandro Sebastianelli, Francesco Mauro and Gianluca Di Cosmo; Supervision, Silvia Liberata Ullo; Validation, Alessandro Sebastianelli, Francesco Mauro and Gianluca Di Cosmo; Visualization, Silvia Liberata Ullo; Writing— original draft, Alessandro Sebastianelli, Francesco Mauro, Gianluca Di Cosmo, Marco Carminati and Silvia Liberata Ullo; Writing—review & editing, Alessandro Sebastianelli, Fabrizio Passarini, Marco Carminati and Silvia Liberata Ullo. All authors have read and agreed to the published version of the manuscript.

**Funding:** This research received no external funding.

**Acknowledgments:** The authors want to thank the editors and the reviewers for sharing their expert opinions on our manuscript, which has benefited from their constructive comments and suggestions.

**Conflicts of Interest:** The authors declare no conflict of interest.

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
