# Peer review of "AIRSENSE-TO-ACT: A Concept Paper for COVID-19 Countermeasures Based on Artificial Intelligence Algorithms and Multi-Source Data Processing"

_ijgi, doi:10.3390/ijgi10010034_

Round 1

Reviewer 1 Report

Dear Authors,

I believe that the implementation of a new tool to support institutions is necessary. Your approach, which consists of using available data is very interesting and deserves to be published in my opinion. As you report, the impact of pollution, specifically air pollution, and Covid-19 spread should be clarified. In this regard, I would suggest adding some references in section 4.2.

As you know, there are several limitations and I would like to see more explanations about them in the discussion section.

I would like to suggest to re-organize the paper moving some part from 2.1 to discussion. I believe that this could be useful for readers.

Author Response

Best regards,

the Authors.

Reviewer 2 Report

This paper presented a description of a new tool to support institutions in the implementation of targeted countermeasures, based on quantitative and multi-scale elements, for the fight and prevention of the COVID-19 pandemic. The article is at a good level and the topic is interesting, I congrats the authors for your effort. However, some aspects should be improved. So, I can recommend accepting the paper after a major revision.

  • The abstract can be rewritten to be more meaningful. The authors should add more details about their final results in the abstract. The abstract should clarify what is exactly proposed (the technical contribution) and how the proposed approach is validated.
  • Overall, the basic background is not introduced well. I recommend the author to extend the Introduction Section by adding more discussion and employing certain intuitive examples.
  • The contributions of the paper are not clearly identified (Section 1, last paragraph). Authors need to be claimed their contributions and justify with sufficient experimental results.
  • The paper does not explain clearly its advantages with respect to the literature: it is not clear what is the novelty and contributions of the proposed work: does it propose a new method? Or does the novelty only consist of the application?
  • Bullet your contribution at the end of the introduction section.
  • Many references are missing (e.g. Section 2 authors should give references for the public databases and ML algorithms), etc. please revise the whole paper carefully.
  • In section 3, the description of the proposed algorithm is not clear, and I strongly suggest using flowcharts or pseudocode to express the principles of this algorithm in order to demonstrate the improvements of the proposed algorithm effectively.
  • Authors need to provide justifications for all the parameters setting.
  • Please highlight the advantages and disadvantages of your method.
  • Analysis parts need to be well written as take-home-messages are not clear. For example, one paragraph - one message.
  • Many recent works that used LSTM with COVID are missing such as:

&& Deploying machine and deep learning models for efficient data-augmented detection of covid-19 infections. Viruses 12, no. 7 (2020): 769.

## A combined deep CNN-LSTM network for the detection of novel coronavirus (COVID-19) using X-ray images. Informatics in Medicine Unlocked 20 (2020): 100412.

^^ Time series forecasting of COVID-19 transmission in Canada using LSTM networks. Chaos, Solitons & Fractals (2020): 109864.

etc…

  • The authors did a poor job with acronyms through the paper. E.g. (DSS) in the Introduction defined many times, etc., please revise whole paper carefully.
  • The paper would benefit much if you ask a native speaker to review and edit the text with a focus on the usage of the English language.

Author Response

Best regards,

the Authors.

Reviewer 3 Report

The described methodology uses a LSTM network for data alalysis. A LSTM is normally a building block of a aNN consisting of many neurons. It would be interesting to see more details of the architecture of the proposed solution.

Input data in a ML project usually requires extensive pre-processing, especially if heterogenous data is used as input. Some words on data pre-processing would be desirable. 

Since COVID has an incubation time of about 14 days, some words on the prediction period time would be interesting. I. e. the delay between the prediction and the actual outbreak.

Some minor orthographic and grammatical errors

Author Response

Best regards,

the Authors.

Reviewer 4 Report

This is an interesting concept paper regarding modeling Covid-19. The authors propose to use neural networks to model the large amounts of data they hope to collect from sensors that collect data regarding air pollution, from environmental variables and from epidemiological data. The discussion is particularly strong in relation to the use of sensors and in the knowledge of particulate matter. The authors carry out an elementary correlation analysis between NO2 and new cases of Covid-19 in both Lombardy and Wuhan.

I have a number of comments.

1. The authors state the time lag between a person becoming infected and testing positive is between 14 and 21 days. This varies with country and testing regime and this long delay no longer applies in most countries.

2. The proposal is very weak regarding the epidemiological issues. No mention for example is made of acquiring data regarding the number of social contacts per person in the population - a key driving force in the Covid-19 epidemic.  There is an absence of references to epidemiology studies in journals like Nature and The New England Journal of Medicine. The authors would benefit from having an epidemiologist on board.

3. The correlation analysis carried out is elementary. The authors state the maximum value of the correlation between NO2 and new cases of COVID-19 is found after 45 days for the Lombardy region, and after 35 days for the Wuhan region. No biological explanation is given as to why the delay is so long.

4. There is an absence of detail regarding what the vector of outcomes Y is the neural net is. The author's speak vaguely of risk, virus diffusion etc. While there is a vast literature on neural networks, mainstream references on the back-propogation algorithm should include names like Rumelhart and Hinton. There is hardly a need to include non-standard references such as arXiv. In addition, there are many well-established studies on time-series forecasting with neural networks. The authors would benefit from having a statistician on board.

5. Overall, the proposal shows some merit but the authors need to demonstrate more familiarity with the epidemiology and neural network literature. 

Author Response

Best regards,

the Authors.

Reviewer 5 Report

This concept paper presents AIRSENSE-TO-ACT project, that aims to create a Decision Support System for the activation of countermeasures during virus pandemics, such the COVID-19. The proposed project is based on a model merging data from heterogeneous sources, including both satellite data and ground wireless networks of low-cost dust monitors. The main ideas of the project are highlighted and case studies regarding the correlation between pollution and Covid-19 are also provided.

Undoubtedly any attempt to address the evolution of the Covid-19 contagion in a systematic and efficient way is of high importance. Even more when it addresses geographic areas like China and Italy where the spread of the virus has been most rapid and emblematic in the first months of the year. While it is noted that it is a concept paper that presents a (potential) project, the aim of this paper is not clear. Initially, a state-of-the-art review should be provided presenting the highlights of similar approaches (not necessary about COVID-19, which is a very recent case, anyway) that fuse information from several sources in order to feed an artificial intelligence system that aims to act as a Decision Support System. Such previous experience, described in relevant recent publications, should be provided in the manuscript as a (short) literature review. The aim of it will be threefold: i) it will present to the interested reader the challenges of the proposed project, ii) it will highlight critical differences between precious cases and the emerge nature of Covid-19 (from a system’s point of view) and (iii) it will act as a reminder of associated difficulties, how they are treated in previous similar attempts and where exactly the contribution of the proposed work relies.

The authors rightly pointed out that, it is beneficial that in this period public databases have further increased in number and type of different available data. Thus, there is a whole bunch of row information that could be eventually used for such a tool. In section 4 an interesting discussion is provided regarding the relationship between pollution and Covid-19 diffusion. However, it confuses the reader since it seems like this feature (pollution) is selected beforehand (or is it the case?). Instead, someone could argue that feature selection should be the outcome of a learning approach, e.g. a deep learning system, that would explore the potential importance of each one of various available features (multi-source data) presented to its input. This is supposed to be the aim of the proposed tool, anyway, and the authors remark it very well as “the need to capture the correlations, sometimes hidden, between this information.”. Thus, the focus should be on describing the characteristics of (at least some of) the various data available and how they are (probably) related.

In the abstract the authors describe the tool as “… a Cloud-based centralized system, single multi-user platform…”. However, no any further information is provided throughout the document regarding these two statements.

At the end of section 3 it is stated that “The final output of the model will be a time series of risk maps,…, with a resolution that can vary from low resolution intended as risk for a country to high resolution intended as risk for a city, or a smaller area.” How is this scaling invariance preserved in the proposed framework? Shouldn't the model be re-trained for each separate resolution since the input parameters (as described in the beginning of paragraph 2.1) are scale depended themselves? If so, doesn’t this limit the applicability of the tool?

It is also not clear what the output of the DSS seems to be. Is it a number indicating how critical the situation is? Or a set of weights indicating the importance of each one of the system’s inputs?

Also, some minor comments:

Just before equation (1) should it be X E R^(n,m) and also Y E R^(p,q) ?

It would be more beneficial to the reader to depict Tables 1 and 2 as figures.

The legends in Figures 10 and 11 are not readable and should be increased in size.

Author Response

Best regards,

the Authors.

Round 2

Reviewer 2 Report

The authors have addressed the reviewer's concerns and the revised version of the manuscript appears to be good.

Reviewer 5 Report

Regarding the comments on the original manuscript, the responses of the authors are enlightening and clarifying. In the revised version of the paper, the questions are systematically addressed and several topics have been significantly improved. The authors have rewritten certain parts of the text, explained critical aspects of the proposed methodology and also enhanced its presentation.